# Characteristics of Pacing Strategies among Elite Cross-Country Skiers According to Final Rank

**DOI:** 10.3390/ijerph19084589

**Published:** 2022-04-11

**Authors:** Vidar Vikestad, Terje Dalen

**Affiliations:** Department of Physical Education and Sport Science, Faculty of Teacher Education and Arts, Nord University, 7600 Levanger, Norway; vidar.vikestad@student.nord.no

**Keywords:** classic style, skating, competition, race strategy, time trial

## Abstract

The purpose of this study is to explore differences in pacing strategies between successful and less successful male elite cross-country skiers during a 15 km interval-start race involving different techniques. The final rank, split times and final times were extracted individually for the top 100 finishers in the 15 km individual time trial races from the Norwegian national season opener races over two years. The same course was used in all the competitions. The athletes were divided into four groups according to final rank: Q1: 1st–25th; Q2: 26th–50th; Q3: 51st–75th; Q4: 76th–100th. The relative change in speed was used for the time spent on lap 1, to an average for laps 2 and 3. Significant correlation between placement and speed reduction after the first lap was found in three out of four races. In Race 2 (skating), both Q1 and Q2 had lower speed decreases between laps than Q4 did. In year 2, both races (classical and skating) had lower speed reduction between laps for the first quartile compared to that of the last. Overall, this study shows that lower-level cross-country skiers started out relatively faster in the first lap and achieved a greater reduction in speed in the subsequent laps when compared to their faster opponents.

## 1. Introduction

In cross-country skiing, the championship program for the Olympics and World Championships consists of a total of six races. Five of these are mass-start races in which the athletes start together, and the first skier to finish is the winner. However, the men’s 15 km race is still an interval time trial, and the athletes are measured by the amount of time taken between the start and finish. This type of race normally takes between 30 and 45 min for well-trained athletes, and this time is significantly affected by the course profile and friction resistance between the skis and the snow. These competitions are usually held on a 5 km circular course on which the athletes complete three laps. Although course profiles vary somewhat, they generally consist of flat, uphill, and downhill sections. In time trial races, the athletes’ tactic is to manage their own intensity in the most efficient way to complete the race in the shortest possible time. Such intensity management is called the athletes’ pacing strategy, referring to a conscious plan for distributing effort, and it has considerable impact on the results of an endurance event [1,2]. The optimal pacing strategy is considered to be the physiologically most effective strategy in a way that all the available energy is used before the end of the competition, but not to the extent that it results in a significant reduction in speed before the finish [3]. Abbiss and Laursen (2008) divided the ways athletes pace different events into six main pacing strategies. They refer to these as: (1) all-out, (2) negative, (3) positive, (4) even, (5) parabolic-shaped and (6) variable pace [4]. Which of these is used will depend on a lot of factors, such as the type of sport, the duration of the competition, the weather conditions, the course profile, the athlete’s tactics and their physical abilities.

Studies of pacing in marathons and race-walking competitions have found that lower-level athletes have a higher starting speed relative to their total race time [3,5]. These two studies examined pacing strategies in mass-start competitions, and it is important to note that several previous studies have shown that having visible competitors can both improve the performance and change the pacing strategy of an athlete [6,7,8]. The physiological effects of different types of pacing were examined during a paced 60 min time trial on a cycle ergometer with both constant and variable intensity [9]. This study found that the test subjects did not experience any additional physiological stress (oxygen consumption, mean heart rate, lactate values in the blood and perceived exertion during the tests) when the intensity was varied by ±5%. This indicates that with minor variations in intensity, one could expect to be able to keep the same average power output compared with an even distribution of intensity. Greater variations in intensity (>10%) appear to cause increased physiological stress and thus lower overall average power output in comparison to steady intensity [10]. 

Mathematical models for dictating optimal pacing strategy have been developed for time trial events in both track cycling [11] and road cycling [12,13]. Attempts have also been made to create mathematical models to show the optimal pacing strategy with respect to cross-country skiing [14]. Such models need to take into account many different variables, and Swain (1997) adds that it would be difficult for cross-country skiers to apply such models during a competition since the athletes do not have direct power-output targets during a skiing race. Swain (1997) also calculated how much time would be saved by following the model, compared to perfectly even intensity, but also pointed out the obvious fact that cross-country skiers do not maintain even intensity during competitions. Therefore, whether or not the pacing strategy used in this model is optimal remains unanswered. Cross-country skiers have fewer options than cyclists for obtaining precise, measurable feedback with respect to intensity management. However, since cross-country skiers know the distance of the laps, they can use the remaining distance to optimize their pacing. Therefore, knowing where or when the finish is may be the most important factor in being able to choose a pacing strategy [15]. Not knowing the course in advance will be relatively rare in interval-start cross-country skiing races, since the athletes usually familiarize themselves with the course in advance of the competition. Furthermore, in the vast majority of cases the athletes have experience from many previous competitions involving similar distances and duration. 

An investigation of cross-country interval-start races showed that the athletes use a positive-pacing strategy, and that those athletes who perform the worst are the ones who start out fastest relative to their own completion times [16]. The same study suggested that weaker skiers should adopt a more even pacing strategy, after having observed how the best skiers had a more even race with less of a reduction in speed over the course of the race [16]. Pacing strategies in cross-country skiing during a 10 km simulated competition with four 2.5 km laps found a 2.4% increase in heart rate between the first and last laps, while the speed was lower during both laps 2 and 3 than during lap 1 [17]. The authors concluded that the athletes were using a parabolic-shaped pacing strategy. With regard to cross-country skiing, different terrain also plays a role with respect to where athletes lose time on a competition course. Welde et al. (2017) examined elite male cross-country skiers and found that lower-performing skiers lose more time on a course than the better skiers do. In addition, they found that speed on the uphill sections drops less between the first and final laps, while the speed on the easier sections falls considerably, and that the speed on the easier sections is the best predictor of their performance [18]. The drop in speed on the easier terrain could probably be explained by increased friction between the skis and snow during the competition, and that this has less impact on the uphill sections where the glide phase is shorter. During a 15 km time trial race, terrain differences are less important when comparing split times of different 5 km laps. Therefore, 15 km time trial races are practical for investigating pacing profiles in cross-country skiers between different similar laps throughout the race. In addition, for the high-performance level of Norwegian cross-country skiers, the national season opener in Norway is well-suited for investigating pacing strategies between different levels of male elite cross-country skiers. An advantage is that these competitions are performed on the same track and with identical rounds year after year. In addition, most of the qualified skiers attend these competitions, since their performance in these races determines if they are given the opportunity to participate in future World Cup and Scandinavian Cup races. Therefore, the purpose of this study is to explore differences in pacing strategies between successful and less successful male elite cross-country skiers during a 15 km interval-start race involving different techniques. 

## 2. Materials and Methods

### 2.1. Experimental Procedures

This study gathered data from 15 km individual time trial races for elite male seniors from the Norwegian national season opener races in Beitostølen over two years. The results were obtained from the open-access webpage, where the official result lists from the races were found (https://www.beitoworldcup.com/tidligere-resultater/, accessed on 1 February 2022). All these races were individual races with three intermediate 5 km laps, and an intermediate time after each lap. The races were FIS races held on the same course with known lap lengths. In total, this study examined two freestyle (skating) and two classic competitions, one from each technique each year. The different techniques are described closer in footnote (https://nordicskilab.com/courses/classic-vs-skate-skiing-which-is-right-for-you/, accessed on 1 February 2022). The final rank, split times and final times were extracted individually for the top 100 finishers in each race. The interval-start races were organized with a one-day break in between, during which a sprint race was held. Many of the athletes, but not all of them, participated in both 15 km races (classical and skating). The start interval between the athletes was 30 s. Based on video analyses, the first lap was adjusted for the longer duration of the two other laps by adding five seconds to the time of the first lap for all the athletes. The study was approved from the Norwegian center for research data (NSD), with reference number 267085.

### 2.2. Participants

The number of participants who completed the races were as follows: Race 1, classical, year 1 (*n* = 156); Race 2, skating, year 1 (*n* = 124); Race 3, classical, year 2 (*n* = 141); Race 4, skating, year 2 (*n* = 133). To eliminate data from athletes whose race had been ruined by a major accident, or from athletes that had more or less gradually given up during the race and had not sought to perform to the best, this study investigated only the 100 fastest skiers from each of the competitions. The qualification requirements for the competitions were set by the Norwegian Ski Federation. The inclusion criteria for participation in the competition was that the 180 (Friday) and 120 (Sunday) best enrolled skiers on the current FIS list could participate, in addition to the top 10 skiers from the Junior Norwegian Cup from the previous year. In this investigation, the 100 fastest skiers from each of the four competitions were included for further analysis (*n* = 400). Written consent from the competitors was not required, as data were in the public domain and no individuals were named.

### 2.3. Course

Information about the course is available on the event’s official webpage (https://www.skiforbundet.no/globalassets/09-rennarrangorer/2021-skan-cup-beitostolen/5km.pdf, accessed on 1 February 2022). The specified length of a lap on the course map is 4700 m, while the official leaderboards show a length of 4900 m. The same course was used in all the competitions. The difference in elevation, from the lowest to the highest points, was 49 m, the lowest point was 787 m above sea level, and the highest point was 836 m above sea level.

### 2.4. Statistical Analysis

To analyze pacing trends and possible differences between different levels of skiers, the athletes were divided into four groups according to final rank: Q1: 1st–25th; Q2: 26th–50th; Q3: 51st–75th; Q4: 76th–100th). Kolmogorov–Smirnov and Shapiro–Wilk tests were used to test the assumption of normality. Between-group (based on final rank) differences in speed reduction were identified with univariate analysis of variance (ANOVA). Post hoc analyses were performed with the least significant difference (LSD) test. Correlation analyses using Pearson’s product-moment correlation were conducted to find the correlation between speed reduction and placement for the 100 best skiers in each race. Relationships were interpreted as follows: r = 0.10 to 0.29, weak; r = 0.30 to 0.49, moderate; r = 0.50 to 1.00, strong [19]. A percentage increase to obtain the relative change in speed was used for the time spent on lap 1, to an average for laps 2 and 3. A *p*-value of <0.05 was considered statistically significant.

## 3. Results

On average, the athletes in 1st–10th place had a race time of 36:10 min, while those in 91st–100th place had a race time of 40:13 min in the 15 km competition. Overall, a significant correlation between placement and speed reduction after the first lap was found in three out of four races. In Race 1, no correlation was found. However, moderate correlation between placement and speed reduction were found in Races 2, 3 and 4 (r = 0.35, r = 0.33 and r = 0.30 (*p* < 0.001 for all), respectively). Total presentation of the individual speed changes between the first versus the average of the second and third laps for the 100 best in the final rank are presented in Figure 1. 

Univariate ANOVA detected no differences in pacing trends between quartiles of skiers according to final rank in the classical race in year 1. However, in Race 2 (skating), both Q1 and Q2 had lower speed decreases between laps (*p* < 0.001 and *p* < 0.01, respectively) than Q4 did. In year 2, both races (classical and skating) had a lower speed reduction between laps for the first quartile (*p* < 0.01 and *p* < 0.05 for classical and skating, respectively) compared to that of the last. Individual data are presented in Figure 1, and differences in percentage change between laps in quartiles based on final rank are presented in Figure 2 and Table 1.

## 4. Discussion

The main finding of this study was that in three out of four races, a moderate but significant correlation between race result and the times for laps 2 and 3 relative to lap 1 existed. This means that the best skiers were those who started slowest relative to their own finishing time, while the skiers who ended up further down the result list started relatively faster in the first lap. The second main finding was that a positive-pacing strategy was used in 393 out of 400 cases in the results that were analyzed. However, the reduction in speed only occurred between the first and second laps, since the second and third laps had the same pace. 

Older athletes may often have more experience from similar competitions and may be able to adapt their opening intensity somewhat better. During the 15 km classic Norwegian Championships, 18 of the preselected, supposedly faster male athletes (top 35/140) were found to have an average age of 26, while the preselected slower athletes (lower half) had an average age of 22 [18]. In contrast to the findings of Welde and colleagues, Losnegard et al. (2016) did not find any group differences in pacing strategy in different levels of female cross-country skiers in the World Cup, World Championships, and Olympic events [16]. The fact that skiers did not ski alone on the course may also have influenced their pacing strategy, since we know that athletes tend to change their pacing strategy if they have competitors around them [7,8]. In the results from our study, there is also reason to suppose that the slowest athletes were more likely to be surrounded by skiers who were faster than themselves. This may cause skiers to adopt a faster opening pace, resulting in a greater reduction in speed during the competition [8]. Losnegard et al. (2022) found that skiers with a pronounced fast-start pattern benefit by using a more even pacing strategy to optimize time trial distance skiing performance [20]. The one race in our study that did not have a correlation was the 15 km classic in year 1, and before this race it had been snowing and there was a lot of new snow on the course. This fresh snow could make the conditions slower, and negatively affect the speed for the early starting skiers [21]. However, it snowed very little during the race. This could have made conditions gradually faster for the supposedly slowest skiers who started first, as the snow in the course changes gradually as the skiers ski on it [21]. Without knowing more about how this might have affected the correlation, we cannot rule out that the weather may have been one explanation as to why we obtained different results from this competition. Moreover, it is very possible that external factors, such as changes in the weather, could have been involved, resulting in different findings to those normally seen in some races. 

Another interesting finding of this study was the positive-pacing strategy used by virtually all the athletes in all the competitions. This pattern has also been found in similar studies on interval-start cross-country skiing races [16,17,18]. Positive-pacing strategies have also been found in several other sports, including rowing [22], swimming [23] and running [24]. However, it is unclear whether or not these findings reflect an optimal pacing strategy. Positive pacing is in contrast with what has been found in, for example, track cycling races [11,25] and world-record marathon runs in recent times [26]. In a 20 km simulated, paced cycle race, an intensity 15% below the self-selected opening intensity during the first 4 min resulted in better completion times [27]. This may be a sign that athletes often overestimate their own abilities at the start of a competition, which can lead to a harder start than optimal. For rowers, the speed over each 500 m section of a 2 km race was 103.3%, 99.0%, 98.3% and 99.7% of the average speed for the entire competition [22], which is very similar to the average percentage speed per lap found on each lap in this study (102.8%, 98.7% and 98.6%). While the starting speed of 800 m on races within 2% of the world record was significantly higher, the speed over the first 200 m was 107.4% of the average speed for the race [24]. On the other hand, this does not necessarily mean that an equal reduction in speed in rowing and cross-country skiing races results in the same reduction in intensity (power output), since the external resistance created by air and friction between skis and snow is different from the resistance against water experienced by a rower. This can make it difficult to compare pacing strategies from different sports. This positive-pacing strategy can probably be explained to some extent by factors other than the pacing of the athletes themselves. One factor is that skis gradually lose their glide during the competition, something which is caused by dirt from the snow that attaches to the structure of the skis and adversely affects the glide [28]. The speed at which skis lose their glide properties can be affected by many different factors, can be difficult to predict or measure and does not necessarily affect each athlete equally, either. To explain why the athletes spent more time on laps 2 (31.3 s) and 3 (32.2 s), compared to lap 1 in the 15 km interval-start races, we can assume that this was largely due to changes in the glide properties of the skis. This may mean that even if the athletes had a relatively significant reduction in speed, they may not have had such a great reduction in intensity (power output) as one might suppose when the lap times are presented. 

The similarity between lap times for rounds one and two (98.7% vs. 98.6% of the average speed for the entire race) shows that the reduction in speed did not follow a linear trend, and that parabolic-shaped pacing was used as the form of pacing strategy. In other words, the speed was the highest in the first lap and relatively similar on the last two laps. A study conducted on well-trained cyclists undertaking two 20 km simulated races in heat and in normal conditions also followed a reverse-J-shaped pacing strategy, where the speed decreased gradually after a fast start, but increased significantly during the final 5% of the tests [29]. Such an increase in speed at the end of a competition was also found in runners over a distance of 10 km [30]. The distinctive character of cross-country skiing, which involves short climbs with rest periods in between, also has an effect on the pacing strategy used. In modern cross-country skiing, 15 km individual-start races are held in and around a stadium area with 5 km laps which often incorporate a certain amount of varying terrain. As a result, it would not be appropriate or possible to maintain a constant intensity, and the athletes adopt a variable-pacing strategy. In other words, they ski with intensity that varies depending on the type of terrain they are in. The course has longer downhill sections where the athletes glide on their skis without actively creating forward propulsion, and this serves as a rest period where the intensity is far lower than otherwise during the race. It is worth noting that when a variable-pacing strategy is examined, the intensity of power output rather than speed or split times is measured (Abbiss and Laursen, 2008). However, this study did not examine pacing between different 1 km distances, but between three different 5 km laps. This excludes differences in terrain, since each lap is performed on the same track. In addition, it is important to note that this track and the laps are familiar to the skiers, since the competition is held on the same track and with the same laps as in a national season opener every year.

Due to the fact that intensity should be measured in watts rather than speed when variable pacing is examined, most studies in this area are carried out using bicycles. By testing athletes during a simulated 10 km cycle race with 1 km segments with inclines varying between 0, 5, 10 and 15%, a study found that a variable-pacing strategy is beneficial compared to steady intensity with the same average power [13]. These results are supported by other investigations into cycle trials [12], as well as by mathematical models [11,31]. From testing simulated time trials under varying wind conditions, it was suggested that the intensity should preferably be measured by using watts, since the heart rate or RPE (rate of perceived exertion) was not sensitive enough to measure relatively small changes in intensity. The mathematical models for dictating the optimal pacing strategy have come a relatively long way for cycling, where a predetermined variable-pacing strategy is also more applicable since the athletes can control their cycling power. However, such models are unlikely to be suitable for use in cross-country skiing because skiers do not currently have a reliable measure for intensity [14]. This means that they must find their optimal pacing by trial and error in competitions and during training. 

Tactical choices based on how other athletes ski their race may also affect an athlete’s pacing strategy during cross-country interval-start races. This is because, unlike in time trial races in cycling, cross-country skiing allows athletes to use the draft behind other skiers. This would be a factor that could affect the pacing strategy of athletes, since skiing behind another athlete can be an advantage. The advantages could be partly due to reduced air resistance created in the slipstream behind an athlete and less friction between one’s skis and the snow on the track behind another athlete. It is also thought that skiing in the slipstream of another athlete could create a psychological advantage, since tailing while cycling has been shown to have a positive psychological impact [32]. In trained competitive cyclists, greater power output, speed and heart rate were apparent during competitor presence in a 16.1 km cycling time trial performance [6]. This study concluded that racing alongside another competitor can serve as an external distraction that can lead to reduced perceived exertion. In this study, it could thus be an advantage to use a different pacing strategy than one otherwise would have used if one were skiing alone, or if skiing in the slipstream of other athletes was not permitted. Consequently, this is an assessment that the skiers themselves must make during the competition. If saving time by skiing behind a faster competitor who is increasing his/her intensity for a period can overcome the disadvantages of a suboptimal physiological pacing strategy, it may be beneficial to use other athletes to control one’s intensity for some periods of the event. 

The results of the present study are limited by the unique characteristics of the Norwegian national season opener races in Beitostølen in terms of race distance and change of elevation. However, this is also a novelty of this study, since most of the athletes were familiar with the laps that are the same year after year. Nevertheless, one should be careful to generalize the findings of the present study to other cross-country skiing races. Furthermore, we analyzed publicly available data without considering other confounding variables such as anthropometry, physiological characteristics and skiing skills. Due to the novelty of the study and the popularity of cross-country skiing in Norway, our findings are of great value for both researchers investigating pacing in sports and coaches working with cross-country skiers.

## 5. Conclusions

This study shows that lower-level cross-country skiers started out relatively harder and achieved a greater reduction in speed when compared to their faster opponents. Although practically all skiers used a reverse-J-shaped positive-pacing strategy, it was still uncertain to what extent any reductions in speed were caused by a reduction in intensity due to factors such as changes in friction resistance between their skis and the snow. Based on our study, it seems plausible to suggest that skiers would benefit from a more even pacing profile in time trial races. In skiing, contrary to running or cycling, the gliding ability of the skis is gradually worsened. Therefore, if skiers have even laps with regard to time, they will actually have a negative split according to intensity. Finally, the data presented provide useful information for cross-country coaches and scientists to describe the characteristic performance profile of 15 km individual time trial races and to design and prescribe specific training programs. 

## Figures and Tables

**Figure 1 ijerph-19-04589-f001:**
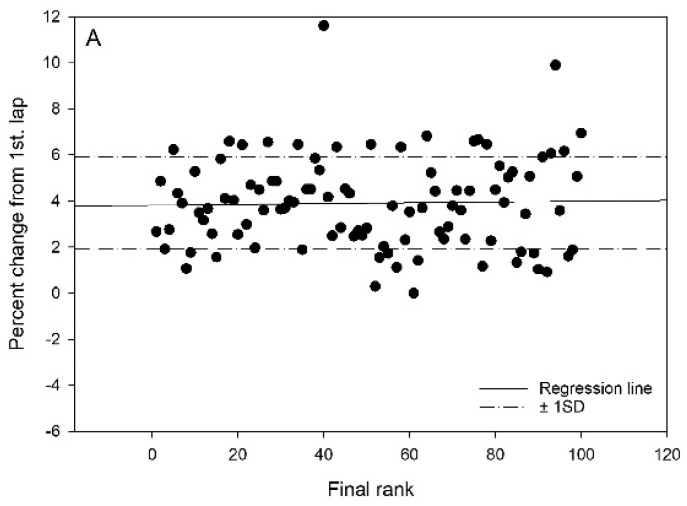
The individual speed changes between 1st versus 2nd and 3rd laps for the 100 best in the final rank in four different cross-country competitions: (**A**) classical, year 1; (**B**) freestyle, year 1; (**C**) classical, year 2; (**D**) freestyle, year 2. Percentage speed changes are calculated as percentage increase/decrease in lap time for an average of the 2nd and 3rd lap, compared to that of the 1st lap. Solid lines represent regression lines, and dash-dot lines represent one standard deviation of the mean percentage change.

**Figure 2 ijerph-19-04589-f002:**
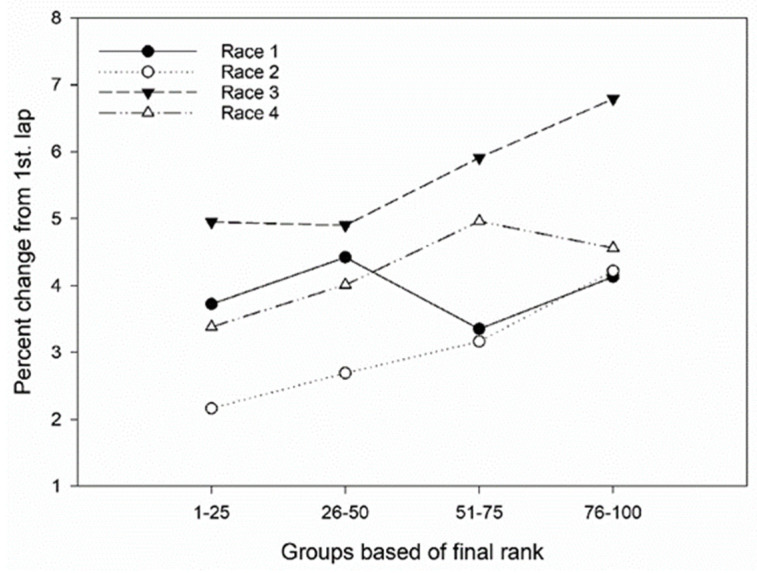
The mean speed changes between 1st versus 2nd and 3rd laps for the different groups among the 100 best in the final rank in four different cross-country competitions: Race 1, year 1 classical; Race 2, year 1 freestyle; Race 3, year 2 classical; Race 4, year 2 freestyle. Percentage speed changes are calculated as percentage increase/decrease in lap time for an average of the 2nd and 3rd lap, compared to that of the 1st lap.

**Table 1 ijerph-19-04589-t001:** Presentation of lap times (seconds) for the top 100 fastest cross-country finishers in the four 15 km individual time trial races from the Norwegian national season opening races in Beitostølen in year 1 and year 2. ^a^ = significantly lower mean speed changes between 1st versus 2nd and 3rd laps than in the group ranked 76th–100th place (*p* < 0.05), ^b^ = significantly lower mean speed changes between 1st versus 2nd and 3rd laps than in the group ranked 51st–76th place (*p* < 0.05), ^c^ = trend towards lower mean speed changes between 1st versus 2nd and 3rd laps than in the group ranked 51st–75th place (*p* < 0.10).

	Lap	Mean ± SD	Min–Max	Rank 1–25	Rank 26–50	Rank 51–75	Rank 76–100
Race 1, classical year 1	1st Lap	824 ± 29	762–877	788 ± 14	812 ± 14	840 ± 11	855 ± 14
	2nd Lap	856 ± 31	785–918	817 ± 13	847 ± 9	868 ± 13	894 ± 10
	3rd Lap	855 ± 28	790–921	817 ± 15	849 ± 8	869 ± 12	886 ± 12
Race 2, freestyle year 1	1st Lap	702 ± 19	658–737	678 ± 11 ^a,c^	701 ± 10 ^a^	709 ± 11	720 ± 13
	2nd Lap	724 ± 24	662–772	693 ± 15	721 ± 7	734 ± 8	752 ± 11
	3rd Lap	722 ± 24	662–777	692 ± 17	719 ± 11	729 ± 9	748 ± 13
Race 3, classical year 2	1st Lap	765 ± 23	703–817	736 ± 15 ^a^	764 ± 9 ^a^	775 ± 13	785 ± 15
	2nd Lap	806 ± 27	731–851	771 ± 19	801 ± 13	816 ± 7	836 ± 7
	3rd Lap	810 ± 28	739–872	774 ± 18	802 ± 10	825 ± 14	840 ± 12
Race 4, freestyle year 2	1st Lap	714 ± 22	662–759	687 ± 11 ^a,b^	709 ± 13	723 ± 8	739 ± 9
	2nd Lap	743 ± 26	673–786	709 ± 14	735 ± 9	758 ± 8	771 ± 8
	3rd Lap	746 ± 27	681–797	711 ± 16	740 ± 11	759 ± 11	774 ± 11

## Data Availability

Data supporting reported results can be found in the including links to publicly archived datasets.

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
