# Peer review of "Characteristics of Pacing Strategies among Elite Cross-Country Skiers According to Final Rank"

_ijerph, 2022, doi:10.3390/ijerph19084589_

Round 1

Reviewer 1 Report

I really appreciate the novelty of the work. Describing pacing strategy in various endurance sports is much needed and also very interesting. I'm very pleased to do a review of this manuscript. The paper is well written. Congratulations to the Authors. I have only few comments. 

Materials and Methods

Participants should be described in more detail (for example the average age). Stats analysis it is adequate, but other variables could be taken into account, e.g. snow structure, weather, route profile.

Results

Very short part of work, in my opinion, this should be described in more detail. Regardless of the above comment - I appreciate appropriate figures and the table.

Discussion

A very well written and also very interesting discussion. What about limitations of the study? It should be included.

Conclusions

I expected a few more conclusions..

Author Response

Dear reviewer,

Thank you for letting us revise our manuscript. Below you can find responses to your comments (responses for the enclosed pdf-file are uploaded).

We thank you for helping us get our paper better and do hope you find the present version of the manuscript acceptable for publication. 

Comment #1: Participants should be described in more detail (for example the average age). Stats analysis it is adequate, but other variables could be taken into account, e.g. snow structure, weather, route profile.

Response #1: We added that the investigation was made in “men elite senior” in the experimental procedures. Our opinion is that the level of the participants are a better description of the participants than the average age. As described in the method section they were elite athletes that was among the 180 (Friday) and 120 (Sunday) best enrolled skiers on the current FIS list in order to participate. Also, the top 10 skiers from the Junior Norwegian Cup from the previous year had opportunity to start. As the introduction displays, the performance level of the Norwegian cross-country skiers are high. As an example of the high levels of athletes, there were on average in the investigated races about 4 minutes differences in race time between 1st-10th and 91st-100th place. In comparison, in World Cup (15 km, Oberstorf 2021) there were over 8 minutes between 1st-10th and 91st-100th place (about 4 minutes difference between 1st-10th and 60th-70th place). We don’t have any official weather data readings, however these races are shown in national television with information about the weather, snowfall and temperature. We have discussed this a bit in the discussion were the weather and snow differed within one competition. Course map with contour lines are in footnote 3 and would work to show the route profile.

Comment #2:Results: Very short part of work, in my opinion, this should be described in more detail. Regardless of the above comment - I appreciate appropriate figures and the table.

Response #2: We agree that the result part might seem a bit short in text, however there are a lot of information in the figures and tables. Therefore, based on the investigation in this study, we struggle to find what other results we should have added.

Comment #3: A very well written and also very interesting discussion. What about limitations of the study? It should be included.

Response #3: We agree, and have therefore added a limitation section in the revised manuscript.

Comment #4: I expected a few more conclusion

Response #4: We agree, we have tried to add some conclusion without being too speculative.

Reviewer 2 Report

12 March 2022

Dear Authors,

I am glad that I could review your work.
In my opinion the topic is quite interesting. The aim of the study is satisfied. Your work should be improved by adding some practical conclusion, recommendation.

Some part of the manuscript should be developed. Please, show that Authors’ study is something more than data analysis.

I put comments in the file that I enclosed to the system. Please, use blue color fond, if you change the text.

Thank you again for your work.

Kind regards,

Reviewer

Author Response

Dear reviewer,

Thank you for letting us revise our manuscript. Below you can find responses to your comments. We thank you for helping us get our paper better and do hope you find the present version of the manuscript acceptable for publication.

Comment #1: I am glad that I could review your work.
In my opinion the topic is quite interesting. The aim of the study is satisfied. Your work should be improved by adding some practical conclusion, recommendation.

Response #1: We agree, we have tried to add something here without being too speculative.

Comment #2: Some part of the manuscript should be developed. Please, show that Authors’ study is something more than data analysis.

Response #2: We have now according to the comments from you and the other reviewers tried to develop different part of the manuscript.

Comment #3: I put comments in the file that I enclosed to the system. Please, use blue color fond, if you change the text.

Response #3: Thank you for the inputs in the enclosed file. We have tried to make a point by point response to your comments in the file. Changes according to your comments are made in the revised manuscript with track changes.

Reviewer 3 Report

The main aim of the paper "Characteristics of pacing strategies among elite cross-2 country skiers according to final rank" is to explore differences in pacing strategies between good and less successful male elite cross-country skiers during a 15-km interval start race involving different techniques.

The topic is interesting with great contribution to the practice, to the coaches and to the competitors. I would like to applaud the efforts of the authors. There are just a few details that need to be explained.

Minor comments:

As correctly stated on line 42, the tempo strategy is also dependent on external conditions including weather, this affects the lubrication of the skis. Was the weather (temperature, snowfall, ....) monitored during the individual races?

There are only 4 sources from the last 5 years in the list of used literature, I recommend adding the recent literature.

Author Response

Dear reviewer,

Thank you for letting us revise our manuscript. Below you can find responses to your comments. We thank you for helping us get our paper better and do hope you find the present version of the manuscript acceptable for publication. 

Comment #1: As correctly stated on line 42, the tempo strategy is also dependent on external conditions including weather, this affects the lubrication of the skis. Was the weather (temperature, snowfall, ....) monitored during the individual races?

Response #1: We don’t have any official weather data readings, however these races are shown in national television with information about the weather, snowfall and temperature …

Comment #2: There are only 4 sources from the last 5 years in the list of used literature, I recommend adding the recent literature.

Response #2: One more relevant study from 2022 was included:

Losnegard, T., Tosterud, O. K., Kjeldsen, K., Olstad, Ø., & Kocbach, J. (2022). Cross-Country Skiers With a Fast-Start Pacing Pattern Increase Time-Trial Performance by Use of a More Even Pacing Strategy. International journal of sports physiology and performance, 1–9. Advance online publication. https://doi.org/10.1123/ijspp.2021-0394

Reviewer 4 Report

Dear authors

I thoroughly enjoyed reading your article and found you observations and comparisons to other sports very insightful. Pacing is a very tricky thing to get right, as can be seen by your findings.

I have no major issues with your work, but have made some suggestions on phrasing below that I believe would improve the article.

Abstract:

Overall, I find the abstract clear, but perhaps ‘good and less successful’ and ‘relatively harder’ could be re-phrased?

For example, successful and less successful.

Maybe ‘relatively faster’ as you refer to a reduction in speed thereafter.

Also, I think you need to explain the format of the race and difference between classical and skating briefly as not all readers will know.

Introduction:

Line 49: note instead of ‘notice’

Line 73-78 I’m a bit confused by this. You state that they familiarise themselves with the course and thus know when / where the finish is, but then state that ‘this actual scenario’ will be rare’. Do you mean that it is rare that they wouldn’t know where the finish is? If so, please amend your writing to make this clearer.

Lines 60-78 you refer to the ‘authors’ repeatedly, but later in you work name the author by name. I think using their actual names first throughout makes for easier reading (e.g. line 89 Welde et al.)

Line 81 I’m not sure if ‘harder’ is the same as ‘faster’ in this context. I think as your are measuring speed / time and not a physiological variable it would need to be ‘faster’. I appreciate going out harder also means you’ll be going faster, but I think in this context it should be re-phrased.

Line 89 what did Wede at al. examine? And sorry, but isn’t it obvious that less successful skiers lose more time than the better ones? Unless you are referring to a specific course, which you do not state.

Line 107 please use ‘successful and less successful ‘ instead of good.

Methods:

A brief explanation between classical and skating techniques should be added here.

Could you add a diagram of the course and of the different skis?

Results:

Link 2 does not work.

Figure 1 is quite small. If permitted, please increase the sizes similar to fig.2.

Table 1. does not indicate the unit. I assume it’s seconds. Please add.

Discussion:

Line 216 onwards: this makes sense, but have you got any evidence that fresh snow is faster? And that those who start first have an advantage?

Line 233 in cycling it has been found that negative pacing is often employed by the more successful riders and that this takes confidence and experience. I do not have the source at hand, but it might be worth finding and including if possible as positive pacing is usually perceived as sub-optimal compared to even and negative pacing.

I enjoyed the discussion and the different points you made. Comparisons with cycling are difficult, but the points about drafting and psychological advantages are clear.

Maybe add a sentence about your recommendations, as the practical application of you work is not clear otherwise.

Author Response

Dear reviewer,

Thank you for letting us revise our manuscript. Below you can find responses to your comments. We thank you for helping us get our paper better and do hope you find the present version of the manuscript acceptable for publication. 

I thoroughly enjoyed reading your article and found you observations and comparisons to other sports very insightful. Pacing is a very tricky thing to get right, as can be seen by your findings.

I have no major issues with your work, but have made some suggestions on phrasing below that I believe would improve the article.

Comment #1: Overall, I find the abstract clear, but perhaps ‘good and less successful’ and ‘relatively harder’ could be re-phrased?  For example, successful and less successful. Maybe ‘relatively faster’ as you refer to a reduction in speed thereafter.

Response #1: We agree and have now changed this according to the comment. 

Comment #2: Also, I think you need to explain the format of the race and difference between classical and skating briefly as not all readers will know.

Response #2: We agree, therefore are the following link put in as footnote in the method section in order to describe the different techniques:  https://nordicskilab.com/courses/classic-vs-skate-skiing-which-is-right-for-you/

Comment #3: Line 49: note instead of ‘notice’

Response #3: We agree and have now changed this according to the comment.

Comment #4: Line 73-78 I’m a bit confused by this. You state that they familiarise themselves with the course and thus know when / where the finish is, but then state that ‘this actual scenario’ will be rare’. Do you mean that it is rare that they wouldn’t know where the finish is? If so, please amend your writing to make this clearer.

Response #4: We agree with this suggestion and have changed the sentence accordingly to clarify what scenario we are talking about. The new sentence is:

Not knowing the course in advance will be relatively rare in interval-start cross-country skiing races, since the athletes usually familiarise themselves with the course in advance of the competition.

Comment #5: Lines 60-78 you refer to the ‘authors’ repeatedly, but later in you work name the author by name. I think using their actual names first throughout makes for easier reading (e.g. line 89 Welde et al.)

Response #5: We agree with this suggestion and have changed the paragraph accordingly.

Comment #6: Line 81 I’m not sure if ‘harder’ is the same as ‘faster’ in this context. I think as your are measuring speed / time and not a physiological variable it would need to be ‘faster’. I appreciate going out harder also means you’ll be going faster, but I think in this context it should be re-phrased.

Response #6: We agree, changed to “…fastest relative to their own completion time”

Comment#7: Line 89 what did Welde at al. examine? And sorry, but isn’t it obvious that less successful skiers lose more time than the better ones? Unless you are referring to a specific course, which you do not state.

Response #7: Changed to:

Welde et al. (2017) examined elite male cross country skiers and…

We agree that it might be obvious that less successful skiers lose more time than better ones, but its their results… We can’t find results from their study that there are differences in the relative change between groups.

 Comment #8: Line 107 please use ‘successful and less successful ‘ instead of good.

 Response #8: We agree, changed according to your comment.

Comment #9: Method: A brief explanation between classical and skating techniques should be added here.

Response #9: We agree, therefore are the following link put in as footnote in the method section in order to describe the different techniques:  https://nordicskilab.com/courses/classic-vs-skate-skiing-which-is-right-for-you/

Comment #10: Could you add a diagram of the course and of the different skis?

Response #10: Course map with contour lines are in footnote number 3, different skies and techniques are described in the link now in footnote 2.

Comment #11: Results: Link 2 does not work.

Response #11: If you copy the text and paste it in as web address, the information will appear. We didn’t make any hyperlink.

Comment #12: Figure 1 is quite small. If permitted, please increase the sizes similar to fig.2.

 Response #12: We agree and have now changed this according to reviewers comment.

Comment #13: Table 1. does not indicate the unit. I assume it’s seconds. Please add.

Response #13: We agree and have now changed this according to reviewers comment. 

Discussion:

Comment #14: Line 216 onwards: this makes sense, but have you got any evidence that fresh snow is faster? And that those who start first have an advantage?

Response #14: The text is implying that fresh snow might be slower, and that it got gradually faster after being raced on. If the reviewer got the exact opposite information from this, are we not being precise enough and have therefore added one sentence (with reference) to better describe this point.

Comment #15: Line 233 in cycling it has been found that negative pacing is often employed by the more successful riders and that this takes confidence and experience. I do not have the source at hand, but it might be worth finding and including if possible as positive pacing is usually perceived as sub-optimal compared to even and negative pacing.

Response #15: We agree, and many of the references included in this study describes even or negative pacing as more suitable.  However, in skiing the skies are losing their glide, so pacing even laps in time, will actually be a negative pacing according to intensity.

Comment #16: I enjoyed the discussion and the different points you made. Comparisons with cycling are difficult, but the points about drafting and psychological advantages are clear.

Response #16: Thank you for positive comments. There are different movement patterns, but still quite similar start to finish style endurance sports.

Comment #17: Maybe add a sentence about your recommendations, as the practical application of you work is not clear otherwise.

Response #17: We agree, we have tried to add something here without being too speculative.

Round 2

Reviewer 2 Report

5th April 2022

Dear Authors,

Thank you for correcting your paper and your comments. I still see some areas to improve the paper.

I leave the decision about some of them to the Editor (e.g. table and column of mean and SD).

Thank you for your work

Kind regards,

Reviewer

Author Response

Dear reviewer,

Our responses to your comments are in the file "ijerph-1644034-review (2).pdf".

Thank you for making us improve our manuscript.